# Amphibole reaction rims record shear during magma ascent

**Paul A. Wallace** [1] ✉, **Janine Birnbaum** [1,8], **Sarah H. De Angelis** [2,8],
**Elisabetta Mariani** [3], **Jessica Larsen**[4], **Jackie E. Kendrick**[1],
**Thomas E. Christopher**[5,6], **Paul D. Cole** [7], **Anthony Lamur**[1] & **Yan Lavallée**[1]

Mineral reaction textures are fundamental archives of geological change. Amphibole reaction rims are among the most widely used to reconstruct pre-eruptive magmatic conditions, traditionally interpreted through changes in pressure, temperature and melt composition. However, these interpretations have largely overlooked the role of deformation, ubiquitous during magma ascent. Here we show that amphibole breakdown is not only thermo-dynamically sensitive, but also mechanically sensitive. Using electron back-scatter diffraction (EBSD) analyses of experimental and natural samples, combined with numerical simulations of crystal rotation under magma flow, we demonstrate that pyroxene nucleates topotactically on amphibole, form-ing rims, but can later reorient in response to strain. In static experiments, gravitational settling alone produces measurable misorientations that can be tracked over time, while natural samples reveal signatures of externally imposed shear. The resulting rim textures encode evolving strain histories, with crystal misorientation distributions tracking both total strain and varia-tions in rim crystallisation and/or deformation rates. With EBSD-derived crystal orientations now shown to capture both thermodynamic and mechanical histories, amphibole reaction rims emerge as four-dimensional petrological recorders, sensitive to pressure, temperature, composition and strain (P–T–X–ε), providing a powerful unified framework for reconstructing magma evolution and the mechanics of magma transport.

The transport and evolution of magma in Earth's crust are governed by a complex interplay of chemical, thermal and mechanical factors[1–4]. To reconstruct magmatic pathways, many petrological studies have con-strained the relationships between melt-mineral chemistry and phase stability in pressure, temperature and composition (P–T–X) space[5–10]. These chemical and thermodynamic perspectives have long domi-nated interpretations of magmatic environments[11,12]. However, the

flow-induced strain associated with simple shear and shear localisation during magma transport through dykes and conduits in the crust has rarely been integrated into petrological models[13–18]. This omission stems partly from the common limitations of current experimental petrology setups, which cannot impose shear stresses while main-taining precise control over fluid pressure, temperature and redox conditions. Within this predominantly P–T–X framework, amphibole

[1]Department of Earth and Environmental Sciences, Ludwig-Maximilians-Universität München, Munich, Germany. [2]Tornillo Scientific, Liverpool, UK.
[3]Department of Earth, Ocean and Ecological Sciences, University of Liverpool, Liverpool, UK. [4]Geophysical Institute, University of Alaska Fairbanks, Fairbanks,
AK, USA. [5]Montserrat Volcano Observatory, Flemmings, Montserrat. [6]Seismic Research Centre, The University of the West Indies, St. Augustine, Trinidad and
Tobago. [7]School of Geography, Earth and Environmental Science, University of Plymouth, Plymouth, UK. [8]These authors contributed equally: Janine Birn-
baum, Sarah H. De Angelis. ✉e-mail: P.Wallace@lmu.de

has emerged as one of the most compositionally diverse, sensitive and petrologically informative minerals in silicic magmatic systems. Its propensity to break down into anhydrous mineral phases (termed reaction rims) under modest changes in chemical potential has made it a cornerstone for reconstructing pre-eruptive magmatic conditions[19–23].

In volcanic rocks, amphibole rims consist of a fine-grained crystal aggregate dominated by clinopyroxene, orthopyroxene, plagioclase and Fe–Ti oxides, yet their modal mineral abundances and microstructures vary widely across—and even within—individual volcanic systems (Fig. 1a, Supplementary Figs. 1, 2 and 3). Some reaction rims are dominated by clinopyroxene, whereas others are richer in different phases; texturally, they may contain microlites that range in shape from equant to acicular, and can display densely packed, symmetrically zoned fabrics to more asymmetric arrangements. Experimental studies reproducing these features under controlled conditions have demonstrated that variations in P–T–X, such as episodes of decompression at varying rates[19–21], elevated temperatures at constant pressure[20,22,23], the presence of a $CO_2$-bearing fluid[23], or shifts in the redox state[23], can account for much of the observed diversity. These differences have been used as fingerprints of the specific pathways by which amphibole breaks down and contribute to decoding the architecture of magmatic plumbing systems (Fig. 1b; refs. 20,24–31). In this framework, reaction rims are interpreted as snapshots of chemical disequilibrium, records of the moment when amphibole became unstable. However, this P–T–X-centric approach implicitly assumes that breakdown proceeds exclusively through thermodynamic triggers, overlooking mechanical processes or deformation–reaction coupling, despite the fact that shear and flow has been shown to accelerate crystallisation/reaction kinetics and mass transfer in magmas, analogous to well-known strain-enhanced reaction progress in metamorphic systems[14,16,32–35].

Here we challenge this assumption by introducing crystallographic and mechanical constraints to investigate shear strain as a previously under-recognised contributor to amphibole breakdown rim textures. By combining data from natural and experimental samples with numerical simulations we show that magma flow during

amphibole destabilisation leaves a systematic crystallographic and microstructural imprint. This repositions amphibole rims as sensitive indicators of both chemical and mechanical environments offering a more complete framework for decoding magma transport processes.

## Results and discussion

### Amphibole breakdown is a topotactic reaction

Electron backscatter diffraction (EBSD) analysis reveals crystallographic orientation relationships that are widely used to characterise microstructural properties in crystalline materials[36]. Over the past decade EBSD has also emerged as a powerful tool for assessing pre-eruptive strain conditions in magmatic systems[13,37]. We apply EBSD to both experimental (Fig. 2a–c) and natural (Fig. 2d–o) amphibole reaction rims to assess if they encode any information on thermodynamic triggers and/or the evolving mechanical conditions during and after their formation. Experimentally-induced amphibole reaction rims, formed by heating above their stability field[22,23], generally reveal a consistent early-stage reaction signature: pyroxene is the first and often the only phase to crystallise during amphibole breakdown. Pyroxene, whether clinopyroxene or orthopyroxene, dominates the reaction rim assemblage and persists throughout its growth with the addition of other phases (i.e. plagioclase and Fe–Ti oxides; Fig. 2d–f).

Crucially, EBSD reveals pyroxene (both clinopyroxene and orthopyroxene) often retains crystallographic alignment with the host amphibole, showing systematic orientations along its c-axis and principal lattice planes (e.g. (100), (010) and (001)). This alignment points to a topotactic relationship[38], a structurally coherent transformation in which the pyroxene, a single-chain silicate, inherits the crystallographic structure and directly nucleates and grows through replacement of the amphibole, a double-chain silicate. Such inheritance minimises crystallographic mismatch while preserving a three-dimensional crystallographic relationship across the reaction interface. In contrast, plagioclase, if present, often aligns approximately perpendicular to the amphibole c-axis, forming semi-coherent interfaces that may still minimise interfacial strain (Supplementary Fig. 4).

Despite the early dominance of topotaxy observed in pyroxenes, its preservation in natural rims is highly variable. In the lavas from

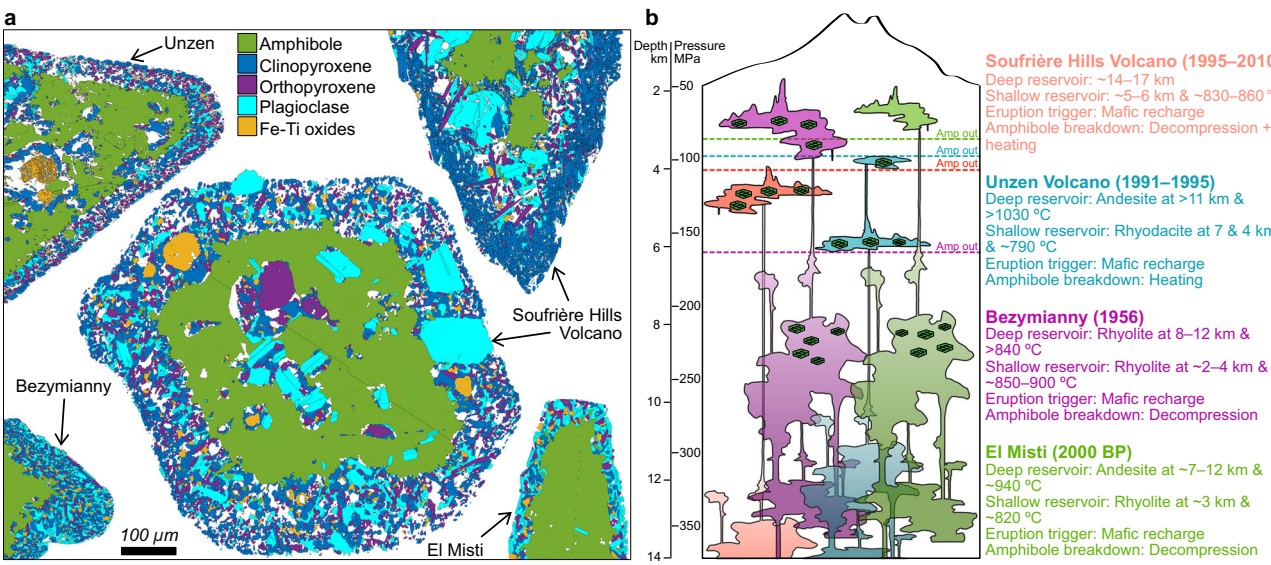

**Fig. 1 | Textural and crystallographic diversity of amphibole reaction rims from four well-characterised silicic volcanic systems: Unzen, Soufrière Hills Volcano, Bezymianny and El Misti. a** Electron backscatter diffraction (EBSD) phase distribution maps of amphiboles and their reaction rims illustrating mineralogical and textural variability both between and within volcanic systems. **b** Schematic representations of the magmatic plumbing systems beneath each volcano showing how

amphiboles have been integrated with other petrological tools to reconstruct complex pre-eruptive magmatic processes. The amphibole symbols represent the depths in the volcanic system where amphibole is reported as stable. Data were compiled from the following sources: Soufrière Hills Volcano[20,39,67], Unzen[30,68,69], Bezymianny[27,28,70] and El Misti[29].

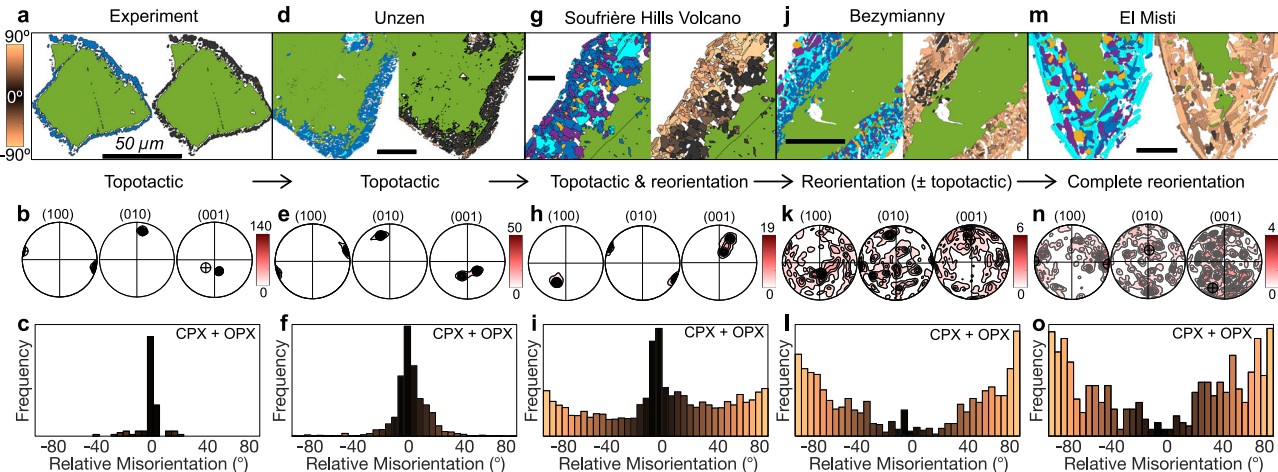

**Fig. 2 | Electron backscatter diffraction (EBSD) phase maps and associated crystallographic orientation data of amphibole reaction rims.** The data represent a representative example of: an experimental sample (**a**–**c**), and natural samples: Unzen (**d**–**f**), Soufrière Hills (**g**–**i**), Bezymianny (**j**–**l**) and El Misti (**m**–**o**). All maps include a 50 μm scale bar. Pole figures depict true misorientations of clinopyroxene rim microlites relative to the amphibole contoured using a kernel half-width of 10° (one point per grain), showing the (100), (010) and (001) planes, with the amphibole orientation indicated by a cross. Colour intensities reflect the multiple of uniform distribution (MUD) with higher values denoting stronger crystallographic preferred orientation. Corresponding histograms display the angular misorientation of combined clinopyroxene (CPX) and orthopyroxene (OPX) populations relative to the amphibole. The colours of the reaction rim grains in the EBSD maps (right-hand of **a**, **d**, **g**, **j**, **m**) and the corresponding histograms represent the angular deviation from topotactic alignment with the parent amphibole, ranging from black for strongly aligned grains to progressively lighter brown colours with increasing misorientation.

Unzen volcano (1991–1995 eruption), amphibole rims display near-zero misorientations between parent amphibole and pyroxenes (Fig. 2d–f), with topotactic relationships maintained even in thicker rims. Similarly, samples from Soufrière Hills Volcano often preserve inner zones with strong topotactic alignment, although these signatures tend to degrade outward, indicating a progressive departure from coherence during rim growth (Fig. 2g–i). Amphibole rim formation at both systems has been associated with pre-eruptive heating following magma recharge[13,30,39], though decompression has also been invoked as the main trigger at Soufrière Hills Volcano[20,40,41].

In contrast, amphibole rims from Bezymianny (Fig. 2j–l) and El Misti (Fig. 2m–o) volcanoes display minimal traces of topotactic inheritance. Instead, pyroxenes in these rims exhibit broad, multi-peaked misorientation distributions, signatures of post-nucleation crystal reorientation or overgrowth. These textures are commonly linked to decompression-driven breakdown[27–29], possibly overprinted by episodic reheating in the case of Bezymianny[31].

Misorientation data confirms a strong topotaxy between amphibole and rim pyroxenes and can serve as a robust metric for comparing amphibole breakdown textures. However, to fully resolve the nature of this relationship, whether it reflects true directional alignment (e.g. [001]) and/or alignment along specific crystallographic planes (e.g. (100)), requires deeper crystallographic analysis beyond misorientation angles alone (see Supplementary information). In strongly topotactic rims (e.g. Soufrière Hills Volcano; Supplementary Figs. 6–11), orthopyroxene grows with (100), (010) and [001] (i.e. c-axes) coincident with amphibole, while clinopyroxene aligns [001] with amphibole but is rotated 180° around [001], producing a mirror-like relationship across (100), a known twin plane in both minerals[42]. In both pyroxenes, [010] (i.e. b-axes) serves as the dominant rotation axis during reorientation by rigid-body rotation. Even in more dispersed rims (e.g. El Misti; Supplementary Figs. 12–15), evidence of topotactic inheritance persists through clustering of clinopyroxene and orthopyroxene [100] axes (i.e. a-axes) around an orientation matching amphibole [100]. Additionally, clinopyroxene and orthopyroxene [001] axes form a girdle within the YZ plane in pole figure plots, suggesting crystal rotation that nonetheless retains crystallographic control. Internal misorientations within rim pyroxenes are generally small (typically ≤3°, Supplementary Figs. 6–11) suggesting that intracrystalline distortion is minimal and that grain-scale reorientation is primarily controlled by rigid-body rotation.

To assess whether deformation by magma flow could disrupt topotactic alignment, we used forward models to simulate crystal reorientation due to rigid-body rotation under strain fields induced by: (1) gravitational settling of the amphibole, (2) flow over a cavity and (3) applied simple shear. Synthetic slices through these model rims reproduce the misorientation patterns observed in EBSD datasets from both experimental and natural samples. The match confirms that even low-to-moderate (~1–10 m/m) shear can erode initial crystallographic coherence, overprinting topotactic textures and broadening misorientation distributions (Figs. 3, 4).

## Crystal settling in petrological experiments

Even in the absence of externally imposed shear, amphibole breakdown in viscous melts does not occur in a truly static environment. During petrological experiments conducted without applied shear, gravitational settling of amphibole crystals generates local flow fields that may influence rim development[43,44]. Amphibole is considerably denser than typical silicate melts and, under experimental conditions, Stokes' law predicts that even slow settling velocities can produce velocity gradients in the surrounding melt[45]. Although most experimental amphibole reaction rims record evidence of a topotactic reaction, local deviations to this are notably observed in higher temperature runs (where melt viscosity would have been lower) or longer duration experiments (Fig. 3a, b). These local deviations in misorientation are most apparent in asymmetric rims, where the thicker portions typically contain a greater abundance of misoriented crystals than the thinner parts, attributed to the gradual outward migration and rotation of rim microlites driven by localised strain fields as the amphibole settles.

Time-series experiments[22], where amphibole grains (~5 vol.%) in a melt were heated 30 °C above their stability field (from 870 to 900 °C) and held over durations ranging from 3 to 48 h, illustrate the reorientation evolution of microlites in the reaction rim (Fig. 3g). In the early stages (3–12 h), the c-axis of pyroxene microlites remain closely aligned with the c-axis of the amphibole, reflecting dominant topotactic nucleation and growth with minimal reorientation. Beyond 24 h, however, misorientation angles broaden, with some microlites

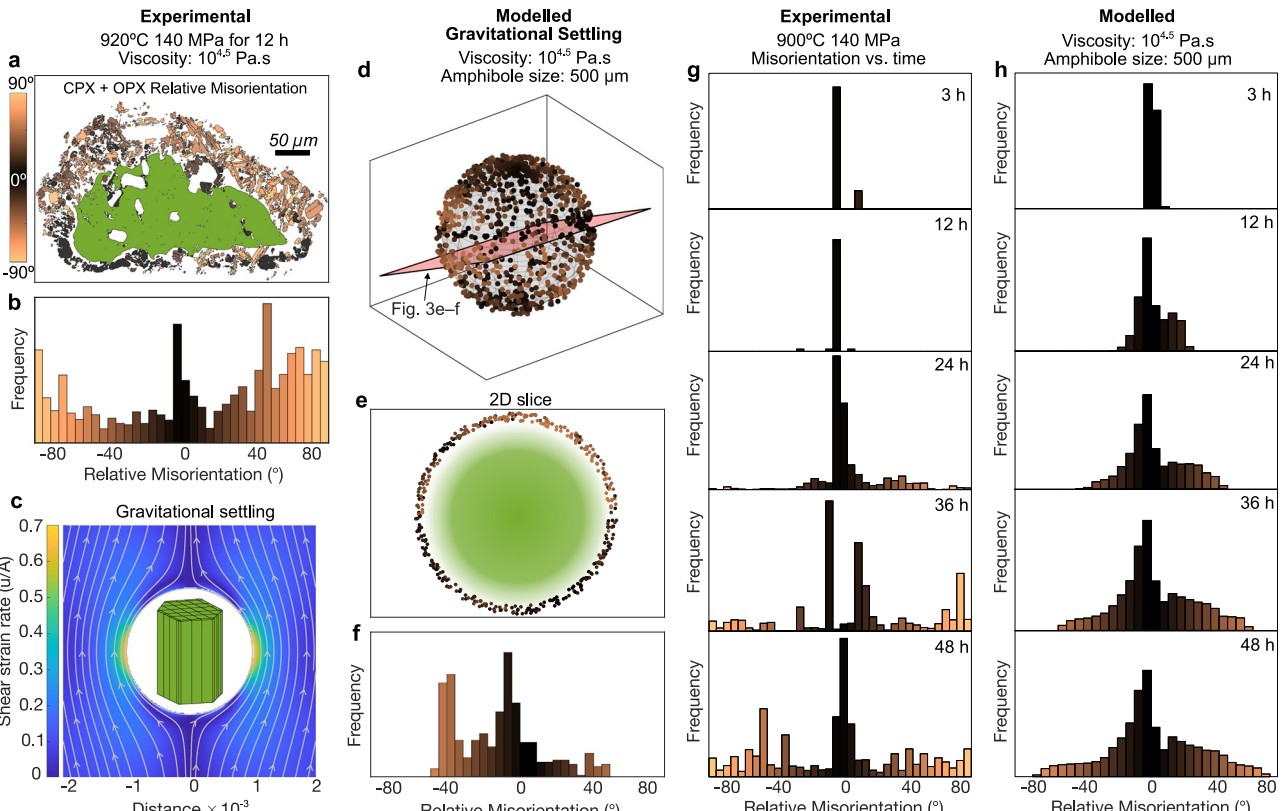

**Fig. 3 | Gravitational settling influences the development of petrological textures in experimental amphibole reaction rims. a** Relative misorientation phase map of an experimentally produced amphibole reaction rim formed by heating outside the amphibole stability field[22]. Colours represent the symmetrised angular misorientation of pyroxene grains relative to the host amphibole (green) quantified in the misorientation distribution shown in (**b**). **c** Schematic illustration of amphibole settling in a viscous melt modelled using Stokes' law, highlighting localised strain concentration at the amphibole–melt interface. Strain is dimensionless and expressed as $u/A$, where $u$ is the characteristic velocity and $A$ is the crystal radius; this scaling yields a strain field that depends only on the scaled radius rather than absolute size. The schematic amphibole orientation is arbitrary and shown for illustrative purposes only. **d** 3D output of the gravitational settling simulation showing that pyroxenes which were initially crystallographically aligned with the amphibole (c-axis vertical), become misoriented due to strain. **e** A selected 2D slice (red) through the 3D model produces a misorientation distribution (**f**) comparable to the experimental data in (**b**). **g** Time-series experiments show a progressive increase in high-angle pyroxene misorientations from 3 to 48 h. **h** Forward modelling of amphibole settling with its c-axis vertical (see Supplementary Fig. 5 for c-axis horizontal) under the same experimental conditions replicates the increasing misorientation trends supporting a mechanical origin for the observed crystallographic variability in (**g**).

displaying deviations of 50–90°, indicating progressive rotation during rim growth. This trend is consistent with the gradual accumulation of strain in the melt surrounding the settling amphibole, which causes microlites to detach and reorientate over time.

These observations can be reproduced by numerical simulations (Fig. 3h) in which we modelled the strain field around a settling amphibole crystal[45] and tracked the rotation of pyroxene microlites that had nucleated during the breakdown process[46] (Fig. 3c). When we simulate microlite nucleation within these synthetic velocity fields and generate 3D misorientation distributions (Fig. 3d), resulting 2D slices (Fig. 3e, f) closely resemble EBSD misorientation distributions from our experiments (Fig. 3a), which are sensitive to the direction of the amphibole c-axis with respect to gravity (Supplementary Fig. 5). These results demonstrate that even weak deformation fields generated by crystal settling can overprint initial topotactic textures, producing systematic, predictable patterns of crystal reorientation.

### Simple shear fields predict natural observations

While experimental studies provide controlled insights into amphibole breakdown, natural samples reveal the imprint of more complex deformation regimes. Many natural amphibole rims formed under dynamic, evolving pressure-temperature-composition-time (P–T–X–t) conditions display crystallographic signatures that deviate markedly from those observed in petrological experiments. These include high-

angle misorientations, apparently asymmetrical textures, and localised heterogeneities in microlite alignment, all of which suggest exposure to sustained or spatially variable shear deformation under stress, beyond that of a simple crystal settling model.

To assess this, we investigated the influence of an externally applied shear stress using two common deformation scenarios relevant to magmatic ascent: simple shear (Fig. 4a–c) and flow over a cavity. In both cases, amphibole rims record strong evidence of mechanically-induced crystal reorientation. Misorientation patterns in rims that formed along amphibole crystal margins and internal fractures (Fig. 4d) are particularly diagnostic. Under shear, only microlites in direct contact with the melt show significant reorientation, while regions shielded from the surrounding flow, such as the interior of fractured crystals, retain the topotactic alignment characteristic of undisturbed breakdown. This spatially confined reorientation pattern was numerically modelled (Fig. 4e) and provides compelling evidence for shear-induced overprinting during rim formation. This same principle applies to reactions observed in amphibole-hosted melt inclusions (Fig. 4h), which remain topotactically aligned owing to their mechanical isolation.

Even in rims showing high-angle misorientations, pockets of topotactically aligned pyroxenes often coexist within the same rim (Fig. 4f, g). Our models reproduce this texture, predicting patchy regions of low misorientation where the shear field is already well-

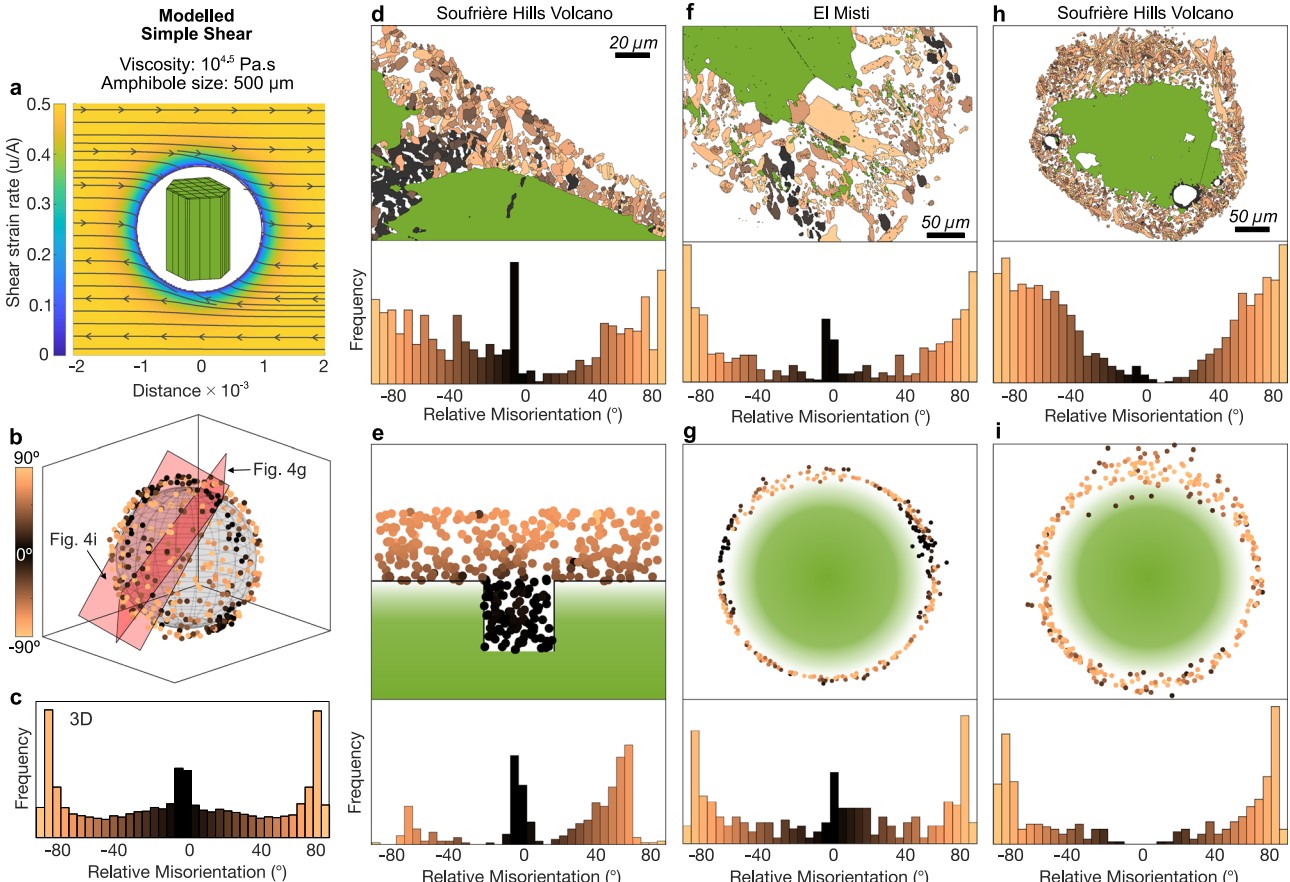

**Fig. 4 | A simple shear model predicts crystallographic orientations in natural amphibole reaction rims. a** Schematic illustration of an amphibole crystal subjected to a modelled simple shear field in which pyroxene microlites experience local strain rate variations. **b** 3D output of the shear simulation showing the development of a spectrum of misoriented pyroxenes quantified in the misorientation distribution in (**c**). **d** Natural example from Soufrière Hills Volcano illustrating variable pyroxene misorientations within amphibole reaction rims comparing structurally heterogeneous regions (e.g. cavities/fractures) to outer edges in contact with interstitial melt, and the resulting misorientation distribution. **e** Numerical simulation of flow over a cavity reproduces the misorientation patterns observed in the natural example shown in (**d**). **f** Amphibole reaction rim from El Misti volcano showing discrete pockets of topotactically aligned pyroxenes surrounded by highly misoriented grains. Such patterns are also observed in 2D slices of the shear simulation (**g**). **h** Example from Soufrière Hills Volcano displaying an apparently asymmetric rim texture with preferential stretching and thickening of the rim toward the upper margin. **i** Model output of amphibole breakdown under simple shear predicts the asymmetric rim through crystal rotation displacement and transport of pyroxene microlites.

aligned with the crystal orientation. In some cases we observe additional signatures of crystal displacement (in addition to rotation) within the rim. For example in Fig. 4h, a slice through a natural amphibole reveals an apparently asymmetrical migration of pyroxene microlites away from the host, consistent with lateral crystal transport under flow as predicted by our model (Fig. 4i). Such a scenario has a measurable impact on the thickness of the reaction rim, a textural feature previously employed as a proxy for magma decompression and thus ascent rate. When rim crystals are transported too far from the amphibole boundary they become entrained in the surrounding shear flow and may ultimately be stripped away or mechanically separated from the host amphibole altogether.

Together, these observations indicate that shear is not only capable of disrupting topotactic relationships but may also play a key role in promoting microlite detachment, spatial segregation and reordering. The transition from topotactic growth to mechanically reorganised textures marks a critical threshold in rim development—where chemical instability alone no longer governs mineral textures, and deformation becomes an active agent in shaping the reaction rim.

### Crystal orientations record shear during magma ascent

In dynamic volcanic systems neither crystallisation rates nor deformation regimes remain constant[47,48]. During magma ascent, stresses, strains and strain rates can rapidly evolve, temporally and spatially, driven by changes in conduit geometry, volatile exsolution and variations in melt viscosity. Amphibole breakdown occurring along this ascent path is therefore shaped by a combination of P–T–X triggers and time-dependent mechanical variables.

To simulate these natural conditions we modelled pyroxene crystallisation around the host amphibole under varying shear ($\varepsilon$) and rim growth rates ($G$). Our model affirms misorientation relationships can be controlled by a competition between $\varepsilon$ and $G$, defining two regimes where growth or shear dominates. When $\varepsilon$ decreases over time or $G$ accelerates, most pyroxenes remain aligned with the amphibole c-axis and only a minority show large misorientations (Fig. 5a, b). This produces distributions dominated by near-topotactic alignment. If $\varepsilon$ and $G$ remain roughly constant over time, the misorientation distribution is that of Fig. 5c, regardless of the exact rate values, as long as sufficient total strain accumulates to affect the earliest formed pyroxenes. In contrast, when $\varepsilon$ increases and outpaces $G$, a greater proportion of the pyroxenes will have experienced large total strains, skewing the population towards high degrees of misorientation (Fig. 5d, e). The modelled misorientation populations align closely with those observed in natural samples from different volcanic systems (Fig. 5f–j).

Reaction rim mineralogy and textures (thickness, microlite size/shape) have been previously interpreted to be characteristic of

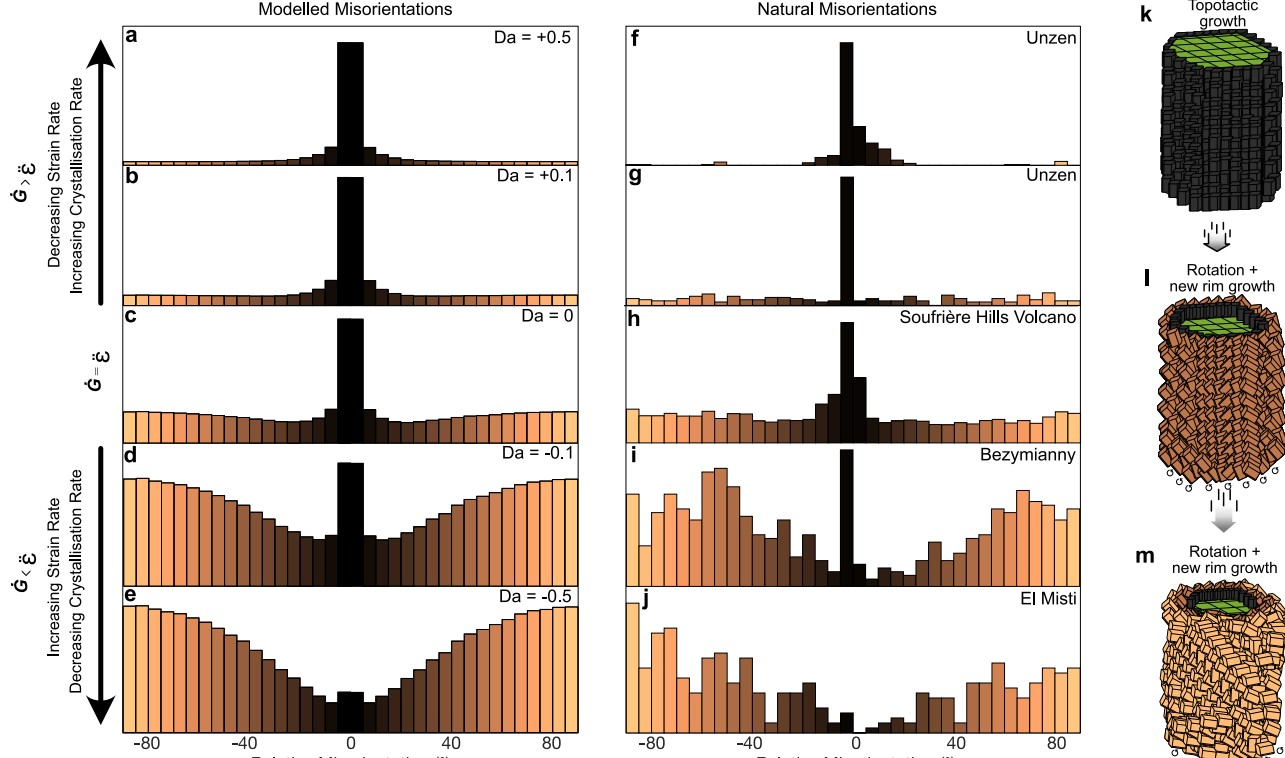

**Fig. 5 | Amphibole reaction rims reflect a competition between crystallisation rate and deformation rate. a–e** Numerical simulations exploring the effect of varying the rate of strain acceleration/deceleration ($\ddot{\varepsilon}$) and amphibole breakdown rates ($\dot{G}$; i.e. pyroxene crystallisation rates) on the misorientation distributions of rim microlites. When growth outpaces deformation (**a, b**) pyroxenes retain topotactic alignment with the host amphibole. Under intermediate conditions where strain rate and breakdown rate are balanced (**c**), rims exhibit dominant topotactic alignment near the amphibole boundary but show increasing high angle misorientations further out. When deformation outpaces growth (**d, e**) pyroxenes lose

topotactic alignment and rotate into high-angle orientations, disrupting the crystallographic continuity with amphibole. **f–j** Misorientation distributions observed in a single natural reaction rim from various volcanic systems capture the full spectrum from strongly aligned to highly misoriented textures. **k–m** Schematic representation of the reaction rim evolution: initial topotactic growth of pyroxenes during amphibole breakdown transitions into rotated, misoriented microlites under increasing shear, while subsequent rim growth continues as amphibole breaks down further.

different breakdown mechanisms (e.g. decompression, heating, volatile fluxing or redox[19–26]). We now demonstrate that the crystallographic relationships within amphibole reaction rims can be attributed to different strain histories through the application of a consistent framework. For example, amphibole rims from Unzen (Fig. 5f, g) and many from Soufrière Hills Volcano (Fig. 5h) have previously been interpreted as a result of heating-induced breakdown[13,30]. These rims preserve dominantly topotactic textures consistent with breakdown under conditions of low-to-moderate $\varepsilon$ and accelerating $G$. In these environments, diffusion-driven rim growth may outpace deformation owing to rising temperatures that accelerate crystallisation by enhancing diffusion at the crystal–melt interface, thus preserving coherent crystallographic relationships even as amphibole destabilises, consistent with a scenario of pre-eruptive heating and/or magma mixing in a weakly deforming magma reservoir (which may be quite localised[49]). The absence of microlite rearrangement after rim breakdown, and ultimately their preservation in eruptive products, may reflect low strain rates during transport, rheological stiffening following amphibole destabilisation, or the low aspect ratios of microlites formed during heating, which are less susceptible to shear-induced reorientation than higher aspect ratio microlites formed during decompression.

In contrast, rims from Bezymianny, El Misti and parts of Soufrière Hills Volcano have been interpreted to record breakdown during decompression-driven ascent[20,27–29]. Crystallographically, these rims exhibit broader misorientation distributions and limited topotaxy (Fig. 5h–j), where the combined effects of increasing $\varepsilon$ and

slowing $G$ foster microlite detachment and reorientation. Such a scenario is expected during decompression-style ascent where crystallisation initiates with volatile exsolution and an attendant rise in viscosity (proportional to diffusion), impeding $G$ over time[50–53]; contemporaneously, decompression inevitably requires large (and likely accelerating) shear strains during breakdown[13,54,55]. Furthermore, crystal-rich domains that develop during magma ascent can facilitate strain localisation due to mechanical interactions between crystals, resulting in heterogeneous strain patterns and allowing pockets of aligned and misaligned microlites to coexist within individual rims. The presence of remnant topotactic zones at the inner edges of some reaction rims (Fig. 2g, j) suggests that amphibole breakdown persisted until shortly before eruption, whereas rims exhibiting minimal topotaxy (Figs. 2o, 5i, j) imply that crystal nucleation halted prematurely, likely owing to a pronounced decline in reaction kinetics prior to eruption. As decompression progresses and volatiles continue to exsolve during the final stages of ascent the increasing melt viscosity drives the system toward the viscous-to-brittle transition, promoting a rheological 'locking in' of the reaction rim texture. In such a scenario the outer portion of the rim, composed of misoriented microlites, may act as an armoured shell (Fig. 6a), mechanically shielding the inner topotactically aligned zone from further deformation. This transition may explain why many rims remain texturally coherent (even after substantial shear) as the stiffening melt progressively inhibits further microlite transport, rotation or disaggregation.

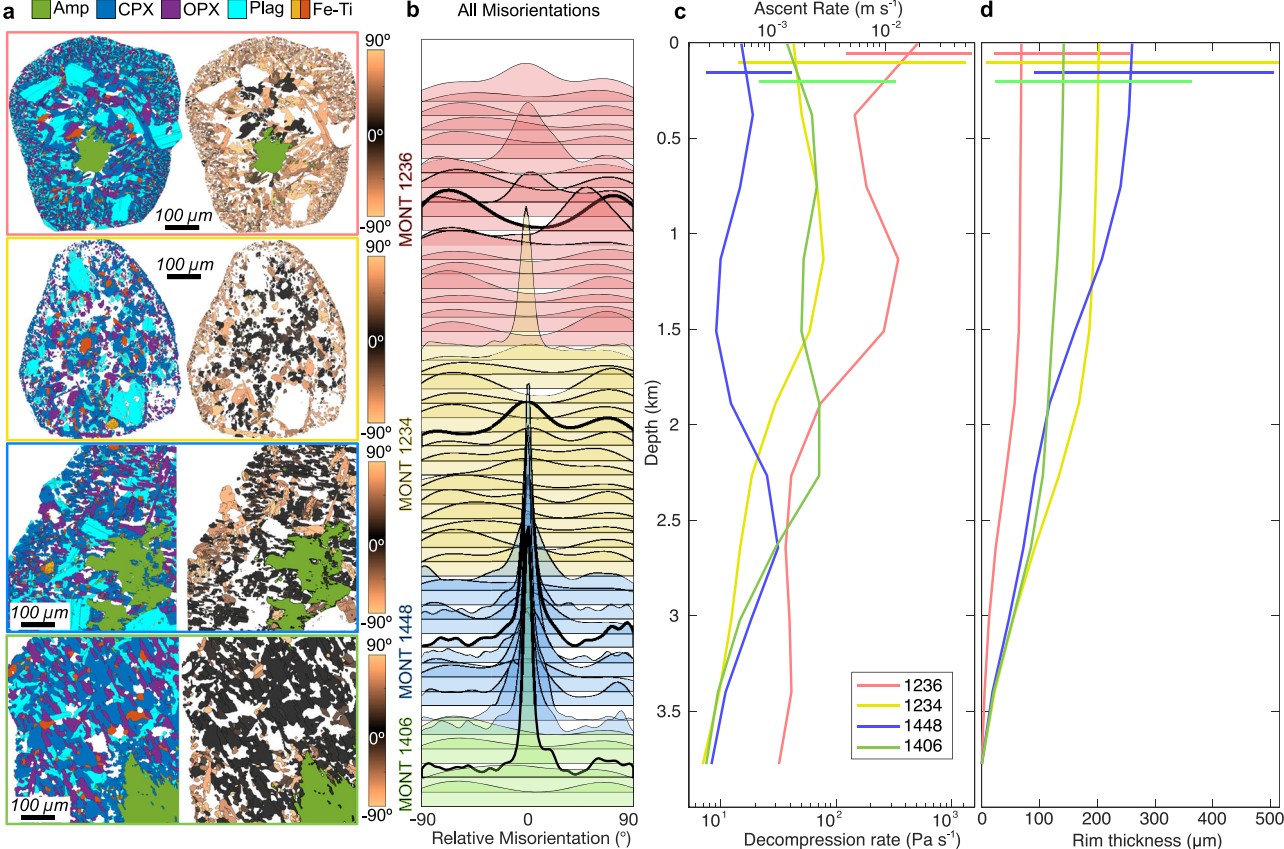

**Fig. 6 | Amphibole reaction rims as quantitative tracers of magma ascent dynamics: an example from Soufrière Hills Volcano. a** Crystallographic misorientation maps from four representative Soufrière Hills Volcano samples. Colours represent mineral phases: amphibole (Amp), clinopyroxene (CPX), orthopyroxene (OPX), plagioclase (Plag) and Fe–Ti oxides. Misorientation maps show angular deviation (0° to ±90°) of pyroxene microlites relative to their host amphibole. **b** Relative misorientation distributions for all reaction rims analysed from Soufrière Hills Volcano plotted as stacked kernel density estimates (KDEs) grouped by sample. Each KDE represents a single amphibole reaction rim and each colour corresponds to a unique sample (labelled), with thick black curves highlighting those shown in (**a**). **c** Monte Carlo-derived ascent models for each sample showing depth-dependent ascent rates and decompression rates calculated from the best-fit misorientation distributions. Colours correspond to the sample labels in (**b**). These profiles suggest variable ascent histories among samples with ascent rates spanning -$10^{-3}$ to >$10^{-2}$ m s$^{-1}$ and decompression rates from -10 to >100 Pa s$^{-1}$. **d** Simulated rim thicknesses as a function of depth for each sample showing how crystal growth interacts with ascent-driven decompression. The variation in rim thickness trajectories reflects the interplay between amphibole breakdown kinetics and strain rate during magma ascent. Horizontal lines in (**c**, **d**) represent the intra-sample variability in rim thickness and propagated uncertainty on ascent rate.

## Reforming current magma ascent models

With EBSD-derived crystal orientations now shown to encode not only thermodynamic but also mechanical histories, amphibole rims emerge as four-dimensional recorders, sensitive to pressure, temperature, composition and strain (P–T–X–ε), thus offering an integrative tool for petrological reconstructions. To illustrate the power of amphibole rim textures as strain markers, we reconstruct magma ascent histories using Monte Carlo simulations on an entire suite of misorientation datasets from four Soufrière Hills Volcano samples (Fig. 6a, b), with each sample containing amphibole reaction rims with a consistent, unique misorientation relationship. These reconstructions demonstrate that amphibole breakdown rims record not only the chemical triggers but also the mechanical context in which breakdown occurs (Fig. 6c, d), including how the evolving balance between reaction rim crystallisation and shear strain rates vary through time. By incorporating strain histories into our interpretations of reaction rims we can reconcile previously conflicting observations of rim texture and mineralogy, and establish a framework in which rim misorientations serve as sensitive strain markers during magma ascent. This revelation broadens the role of amphibole as a crystallographic and textural archive of the evolving interplay between crystallisation, deformation and eruption.

## Methods

### Sample selection and preparation

Reacted amphiboles from both experimental and natural intermediate-silicic volcanic samples were prepared in order to investigate crystallographic orientation relationships between the host amphibole and reaction rim microlites.

Natural samples: we selected samples from volcanic systems for which well-documented interpretations of various amphibole reaction rim triggers are available (Supplementary Fig. 1). Four samples were selected from different deposits of Soufrière Hills Volcano, Montserrat: (1) a crystal-rich andesite from a pyroclastic flow deposit, emplaced in 2002 ($n = 16$ amphibole reaction rims analysed); (2) a crystal-rich andesite from a pyroclastic flow and surge deposit, emplaced in 2003 ($n = 20$); (3) a dense block with a porphyritic texture, emplaced in 2006 ($n = 5$); and (4) a mafic inclusion hosting xenocrystic amphibole, emplaced in 2006 ($n = 9$). Three samples were selected from the 1956 eruption of Bezymianny Volcano, Kamchatka[56], including two andesitic cryptodome samples from the directed blast ($n = 3$ and 11) and a single sample from a pyroclastic flow deposit ($n = 6$). Three samples were selected from the 2000 B.P. eruption of El Misti Volcano, Peru[29]; all three samples were ignimbrite facies pumice from a pyroclastic density current deposit ($n = 4$, 11 and 5). Two samples were selected from the 1991 to 1995 lava dome at Unzen Volcano, Japan[13].

Full sample details are provided in Supplementary Table 1. To ensure the range of amphibole reaction rims selected were representative for each sample, amphiboles were deliberately selected to span the full spectrum of observed rim textures in each sample (see selected amphiboles in Supplementary Fig. 1), including thin rims, thick rims, and nearly completely pseudomorphed grains (see Supplementary Fig. 3). This targeted sampling strategy was designed to capture the crystallographic evolution of amphibole breakdown across different stages of reaction progress. Importantly within each eruption, amphiboles with different rim thicknesses exhibit consistent crystallographic misorientation distributions, indicating that the deformation signal extracted from the rims is robust and not dependent on selecting a particular subset of crystals.

Experimental samples: experimental samples were from the studies of De Angelis et al.[22] and Wallace et al.[23]. All samples were from heating experiments wherein the experimental charge was first equilibrated at relevant amphibole stability conditions, followed by single-step isobaric heating to generate thermal disequilibrium. Experiments conducted by De Angelis et al.[22] were performed at $fO_2$ of Re-ReO (-NNO + 2) with $XH_2O/XCO_2$ of 1.0/0.0. Times series experiments (3–48 h) used in Fig. 2h were heating experiments performed at 900 °C and 140 MPa. The experiment conducted by Wallace et al.[23] was performed at $fO_2$ of -NNO + 1 with $XH_2O/XCO_2$ of 0.7/0.3. Full details of the samples and experimental conditions are provided in Supplementary Table 2.

Sample preparation: thin sections from the natural samples and epoxy-mounted experimental charges underwent standard polishing procedures concluding with a final colloidal silica polish to create surfaces that were suitable for high-quality EBSD. Finally, samples were coated with a thin (~25 nm) layer of carbon.

Amphibole selection: representative regions encompassing the transition from the host amphibole into its reaction rim were selected to ensure that the continuity between host (amphibole) and daughter phases (clinopyroxene, orthopyroxene, plagioclase and Fe–Ti oxides) was captured.

## Electron backscatter diffraction (EBSD) data acquisition and processing

Analytical conditions: highly polished thin sections were analysed by EBSD using a CamScan X500 CrystalProbe field-emission gun scanning electron microscope (SEM) using the AZtec EBSD acquisition software from Oxford Instruments HKL. Analyses were performed using a 20 kV accelerating voltage and 30 nA beam current. The electron beam source column is tilted at 70° relative to the sample surface[57]. Step size was selected adaptively based on the characteristic grain size of each dataset (reaction rim grain size typically ranges 3–20 µm), with finer step sizes (0.1–0.2 µm) used for fine-grained reaction rims and larger step sizes (0.5–1.0 µm) used for coarser-grained rims to ensure adequate spatial resolution for grain reconstruction. Indexing was performed on hornblende, augite, enstatite, anorthite, biotite, magnetite and ilmenite using crystallographic match files with known crystal symmetries[58], sourced from the American Mineralogist database (using best in database family and 47–50 reflectors).

MTEX analysis workflow: all EBSD data were exported from Aztec and imported and processed in MATLAB using the freely available MTEX toolbox (version 6.0.0). The .cpr files were loaded into MTEX with the crystal symmetries for all indexed phases specified. A region of interest was defined interactively to exclude extraneous or noisy data and microlites in the groundmass that do not form part of the reaction rim. The EBSD data within the selected region were then gridified, ensuring a properly spaced grid of orientations suitable for subsequent analyses.

Grain reconstruction: a grain-reconstruction step identified individual grains based on a misorientation threshold of 10° and a minimum of three pixels per grain ensuring the reconstructed grains were

sampled by significantly more than three pixels. Reconstructed grains were optionally smoothed to remove spurious boundary pixels. MTEX's calcGrains function was used to segment grains and assign each measured orientation to a specific grain ID. The average orientation of each grain was calculated via MTEX's meanOrientation method.

Defining the reference hornblende grain: because the goal was to measure the relative difference in misorientation between reaction rim phases and a representative amphibole orientation we identified the largest amphibole grain (i.e. the grain with the greatest two-dimensional area) and extracted its mean orientation as the reference. This orientation denoted OHornblenderef, serves as the crystallographic benchmark.

Computing relative misorientations: for every augite, enstatite and anorthite grain (i) with a mean orientation OAugite(i), OEnstatite(i), OAnorthite(i), the misorientation angle ($\Delta\theta i$) was computed via the MTEX function: $\Delta\theta i = \angle$(OAugite(i), OHornblenderef); $\Delta\theta i = \angle$(OEnstatite(i), OHornblenderef); $\Delta\theta i = \angle$(OAnorthite(i), OHornblenderef), where $\angle$ returns the smallest rotation required to bring one orientation into coincidence with the reference amphibole. MTEX returns the result in radians which we converted to degrees. Where misorientation angles exceeded 90° we folded them back into the 0–90° range by subtracting 180°, ensuring a symmetric distribution about 0°.

Visualisation: the distribution of these relative misorientation angles was visualised as maps by colour-coding individual grains (0° to ±90°), plotted as a histogram and, for comparing whole sample variability, as a kernel density estimate to illustrate how similar (or dissimilar) reaction rim grains were to the reference amphibole orientation. A high frequency near 0° misorientation indicates grains are closely aligned crystallographically to the parent amphibole grain. By contrast a broad or multimodal distribution would imply more varied orientations.

## Numerical modelling methods

We test our hypothesis of shear-driven crystal rotation as a mechanism for misaligning pyroxenes using simulations of the rotation of elongate crystals in a few simple flow geometries that capture the likely dynamics in the experimental and natural samples. We consider three possible flow fields: (1) gravitational settling of the amphiboles with respect to their less dense host melt; (2) flow across a flat surface with a square notch (a variant of the classic lid-driven cavity problem); or (3) rotation in simple shear surrounding the amphibole. These flow fields have well-known analytical (1 and 3; refs. 45,46) or numerical (2; refs. 59,60) solutions. The rotation of elongate ellipsoidal particles was first determined analytically[46] and has been experimentally verified for a variety of elongate particle shapes[61,62] and extended to consider dip with respect to the shear plane[62,63], a combination of pure and simple shear[64], and interactions between crystals[65,66].

We begin by initialising a population of pyroxenes in the region surrounding the amphibole with random locations and a truncated normal distribution of aspect ratios centred about 8 with a standard deviation of 5 which have an initial orientation with their long (c-) axis parallel to the host amphibole. The initial crystallographic orientation of each pyroxene is defined relative to the host amphibole following the topotactic orientation relationships observed in the innermost reaction rims. Misorientation is subsequently calculated as a rigid-body rotation through time as the angular deviation between the evolving pyroxene orientation and the fixed crystallographic reference frame of the amphibole host. We allow the pyroxenes to rotate in 3D using the known expressions[62] and advect passively in the local velocity field using a forward Euler scheme. The resulting 3D solutions are then cut by a 2D plane which shows a potential view that matches the 2D nature of the EBSD analysis. See Supplementary Table 3 for a summary of the model input parameters. For an extended

methodology and Supplementary Figs. 16–23, see online Supplementary information.

## Data availability

The datasets generated in this study that support the findings of the article have been deposited in the Figshare repository at https://doi.org/10.6084/m9.figshare.31451656. All deposited datasets are provided in formats compatible with the MTEX toolbox to facilitate reuse and reproducibility. The raw EBSD acquisition files are available from the corresponding author upon request.

## Code availability

The codes used to simulate shear deformation and generate the modelling results presented in this study have been deposited in the Figshare repository at https://doi.org/10.6084/m9.figshare.31539913. The code used to process the EBSD data has been deposited in the Figshare repository at https://doi.org/10.6084/m9.figshare.31451656.

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

## Acknowledgements

We thank the VUELCO team for their invaluable field assistance during a 2012 sampling campaign at Soufrière Hills Volcano. For Unzen volcano samples, we gratefully acknowledge Takahiro Miwa and Hiroyuki Shimizu for field assistance during a 2016 campaign, along with funding provided by the Daiwa Anglo-Japanese Foundation Award (grant number 11000/11740). Pavel Izbekov is thanked for generously providing samples from Bezymianny volcano (funded by PIRE-Kamchatka, NSF OISE-0530278), and Shanika De Silva for insightful discussions and supplying samples from El Misti volcano. Their contributions were essential to the completion of this study. P.A.W., A.L., J.E.K. and Y.L. acknowledge support from the LMUexcellent fund, funded by the Federal Ministry of Education and Research (BMBF) and the Free State of Bavaria under the Excellence Strategy of the Federal Government and the Länder. J.L. acknowledges a National Science Foundation (NSF) grant no. EAR-1650185. Y.L. and J.B. acknowledge support from the European Research Council (ERC) MODERATE project grant no. 101001065. The EBSD-SEM laboratory at the University of Liverpool (now the scanning electron microscopy shared research facility-SEM SRF) is acknowledged for supporting EBSD analyses.

## Author contributions

P.A.W.: writing-original draft, writing-review and editing, conceptualisation, visualisation, validation, resources, project administration, methodology, investigation, formal analysis. J.B.: writing-review and editing, visualisation, validation, methodology, investigation, formal analysis. S.H.D.A.: writing-review and editing, conceptualisation, project

administration, methodology, investigation, formal analysis, resources. E.M.: writing-review and editing, resources, validation, funding. J.L.: writing-review and editing. J.E.K.: writing-review and editing, resources. T.E.C.: writing-review and editing, investigation, resources. P.D.C.: writing-review and editing, investigation, resources. A.L.: writing-review and editing, investigation. Y.L.: writing-review and editing, funding.

## Funding

## Competing interests

The authors declare no competing interests.

## Additional information

**Supplementary information** The online version contains Supplementary material available at https://doi.org/10.1038/s41467-026-71477-x.

