## [Transparent Peer Review File · Nature Communications]

Amphibole reaction rims record shear during magma ascent

Corresponding Author: Dr Paul Wallace

Version 0:

Reviewer comments:

Reviewer #1

(Remarks to the Author)

Wallace et al present an interesting and convincing investigation into the impact shear deformation on the texture of reaction rims formed during the breakdown of amphiboles during their eruption. It is well known that pressure and temperature changes can quickly result in the breakdown of amphibole when taken outside its stability field, and many workers have used this to infer processes that occurred during historic eruptions. In some cases the texture of the reaction rims has been attributed to decompression and hence ascent rates of the magma can be determined. Such studies are significant in volcanic hazard assessments. In other eruptions, arguments have been made for temperature control on rim formation. A combination of factors was suspected. However, this new work by Wallace et al shows that some of the textural variability is due to shear deformation in the conduit during magma flow, and that the fluid properties can change and potentially be deciphered by micro-imaging. This adds to the potential information on magma ascent that could be inferred from amphibole rim textures. The authors use examples from well-studied volcanic eruptions that are known to show some variation in processes, and compare them to experimentally made crystals. They introduce imaging and crystal orientation techniques that will be useful in future studies.

Hence, the work is noteworthy. Understand the processes of explosive volcanic eruptions is a topic of wide interest. This is an important contribution to the literature on interpreting amphibole rim textures for volcanic phenomena. It is new and compliments the current literature. The conclusions all seem logical from the data presented and the description of the methods is sufficient to be convincing and would act as a basis for using this approach in other studies in the future. There is sufficient information on the numeric or modelling, but I am not an expert on such calculations. However, it is compelling.

Minor text error?: in paragraph 3 and 4 of section named "amphibole breakdown...", the wrong panels in Fig 1 are referred to. Check.

Overall, a welcome contribution, worthy of publication.

Phil Shane

Reviewer #2

(Remarks to the Author)

Summary

This paper presents EBSD-derived crystallographic data from amphibole reaction rims, including phase and crystal orientation data, to investigate the influence of deformation on the nature of amphibole breakdown during magma transport. They find that while some reaction rims record systematic topotactic relationships between the amphibole and pyroxene grains, other rims contain systematically reoriented grains. Models are consistent with this reorientation relating to shear deformation during magma ascent.

The authors' consideration of deformation during magma transport certainly poses an important question about how microstructures are interpreted to reconstruct a P-T-t path. The manuscript presents an interesting dataset spanning multiple volcanic systems and compares natural microstructures with models, but the assumptions used in modeling the deformation are not clearly presented. However, since I do not have any expertise in magma transport and evolution, my review is focused on the interpretation of the crystallographic data and the discussion of deformation. The manuscript would benefit from some refinement of the figures and much more clarity about the style of deformation that seems to be implied throughout the text. I think this paper can be a meaningful contribution, and I recommend major revisions.

Major Comments

From what I can gather in the supplement, the forward modeling of crystal orientations is assuming that the newly formed crystals are freely rotating as rigid particles within a viscous matrix driven by a velocity field. A misorientation is calculated given some initial crystallographic orientation of the amphibole and pyroxene grains. Given that the rotations observed from the EBSD data (i.e., the b-axis is the dominant rotation axis), it seems there could be some contribution of intracrystalline deformation to the reorientation of these grains. Could this contribution be significant, or are there any constitutive relationships that are included in the model setup? Perhaps the strain rates are much too fast for this to be under consideration, but with velocities down to 10^{-9} m/s and temperatures of >800 °C, I think it's a possibility. At the very least, the authors do not provide an adequate interpretation for why the rotations they observe are so systematic in line with the crystal directions during the deformation-induced reorientation. Alternatively, are the microlites consistently anisotropic grains that are also demonstrating a shape-preferred orientation as their long axes align with the flow direction? Overall, there is a significant lack of information when it comes to the interpretation for a mechanism behind reorientation beyond "deformation."

Currently, the main text is not clear about the type of deformation, in terms of common deformation mechanisms, the authors are discussing throughout the manuscript. There is only some clarification in the methods section at the very end of the manuscript. At a minimum, I would ask that some further clarification about the model setup and assumptions is added to the main text of the paper indicating the kind of deformation the authors are modeling. I would also recommend adding a table to the supplement or an Extended Data Table 3 that summarizes the model parameters and the values used.

Minor Comments

Lines 42–43 What kind of deformation is expected (or has been observed) during magma ascent?

Lines 64–67 References for the influence of deformation on reaction kinetics?

Lines 136–137 Is this reorientation driven by rigid grain rotation or dislocation creep?

Lines 163–164 The source of these experiments should include a reference here.

Lines 563–564 The match files used in Aztec must be reported. "Standard files" is insufficient.

Line 572 The suitability of a minimum of 3 pixels per grain for the grain reconstruction can only be determined from the combination of the step size (0.1–1.0 microns) and the grain size. Please report here what the typical grain size range was for the pyroxene grains.

Figures These figures are very dense and not easy to read. Many of the plot labels are much too small. I think most if not all the figures would benefit from adding further differentiation between labels and sublabels as well as more white space between the separate sections.

Figure 1 This is a very dense figure. The labels are hard to read when placed over the data. In the caption, the (100), (010), and (001) planes should still include the parentheses indicating that they are planes.

Additionally, the print resolution of the figure overall, but particularly the orientation maps and pole figures is insufficient for the level of detail, I strongly recommend making sure the final version has a higher dpi.

Figure 1a Why does this map only include one complete grain? Assuming this map is made from grain separates, what's the reason for not making complete maps of these 5 grains? And are these 5 grains different from the grains used in Fig. 1c,f,i,l,o?

I would also recommend listing the 4 volcanoes in the legend on the right-hand side of Fig. 1a in the same order that they appear in Fig. 1c–q.

Figure 1b Please state explicitly in the caption what the amphibole symbols mean in the schematic. Is that depth levels where amphibole was recovered from samples? Are those samples included in the manuscript?

Figure 1c... Please state in the figure caption what half-width was used for contouring these 1-point-per-grain pole figures.

Figure 2c Please clarify the variables used for plotting strain (u / A) in the caption. And is the orientation of the cartoon amphibole grain representative of the initial orientation? Or is everything relative and the actual orientations aren't significant?

Figure 2a,g The source of the experimental samples should be referenced in the figure caption.

Figure 5a Scale bars should not be placed over busy data as they become too difficult to read. I suggest placing a white or transparent box underneath for readability.

Reviewer #3

(Remarks to the Author)

Dear Authors and Editor,

The manuscript entitled "Amphibole Reaction Rims Record Shear during Magma Ascent" by Paul A. Wallace and collaborators, proposes a framework to use amphibole breakdown rims as tools to quantify strain in magmas. With EBSD (Electron backscatter diffraction) measurements on natural data and experimental results to quantify the orientation of

amphibole and the crystals composing their breakdown rims they hypothesize that re-orientation of crystals was the result of deformation. To test this, they use flow mechanics simulation of amphibole surrounded by elongate pyroxene exposed to several deformation fields (crystal settling, flow across a flat surface with a notch, simple shear), and calculate the reorientation of pyroxenes over time. They conclude that in the case where Strain rate decreases relative to crystallization rate (development of the rim), the crystal grows following the alignment of the "host" amphibole (topotactic growth), and that with the opposite situation, misalignment with host amphibole increases.

This paper shows a novel and very interesting approach to use amphibole breakdown rims that could help to quantify to some extent magma deformation during its late storage, ascent and eruption. Data from models, petrological experiments and natural samples seem to agree with the argument made as well and represent a quite extensive EBSD dataset. However, I have some reserves because of the underlying assumptions made during the modelling approach that uses only a single crystal exposed to flow (Stokes flow, simple shear), without considering the influence of multi-crystals, which has been shown to drastically change flow simulation results. Besides, I would have liked to see more data on the breakdown rims from natural products (Percentage of amphiboles with rims, thickness distribution of the rims for instance) to be able to assess the representativity of the sample chosen for EBSD. The conclusion using the MCMC model is also very interesting, but I think it lacks some of the data that were used to calculate the decompression rates using amphibole breakdown rim models.

Overall, I think this study fits in Nature Communications thanks to its novelty and comprehensive approach, but I think there is still some key points that need to be addressed and discussed for this study to be published (see my more detailed comments below).

Best regards,

Olivier Bernard

Main comments:

- Single crystal flow models:

In your model about crystal settling, you use a simple case of Stokes law for slow settling velocities. However, above even relatively small crystal volume fractions of a few percent, this type of settling doesn't completely describe the forces applied on the crystals. For all your case studies, you are above this threshold, and therefore other forces applies slowing down crystal settling such as Basset forces, lubrication forces, hindrance and crystal-crystal interactions making the dynamics a lot more complex and the resulting deformation field very different (see Faroughi and Huber 2015, Culha et al., 2020, Huber et al., 2025 Ann Reviews). This could have a big influence on your modelled misorientation outputs. The dynamics of two-phase flow (crystals + melt) can be very complex and upscaling to a whole magma chamber or conduit the behaviour of a single crystal in an infinite fluid volume is a severe simplification.

Clustering particularly can occur even at low crystal fractions (Culha et al., 2020), and mean a cluster would behave as a single settling particle – could this also explain the presence of asymmetric rims?

I think all these limitations should at least be mentioned and discussed in the manuscript, and maybe some testing could be done on the simpler crystal settling case.

- Sample representativity:

How representative are amphibole analysed for each eruption?

EBSD is a time-consuming method, so I understand that it is not possible to analyse hundreds of amphibole per eruption deposit, but it would be good to have an idea of amphibole size distribution in each eruption: how many have rims, what is the thickness distribution of these rims, etc (in a similar way that you show on your extra figure 1 for rim mineralogy).

Line 222-224: You mention, loss of crystals from the rim because of the strain applied to the amphibole. Did you quantify it? With a given strain, how much of the rim thickness could be lost?

It would be nice to have details on the parameters used after the MCMC simulations to obtain decompression rates at each step from the amphibole breakdown rim models.

Minor Comments :

Line 98: it's Fig 1f-o

Line 541-546: inconsistency with the cited study.

Version 1:

Reviewer comments:

Reviewer #2

(Remarks to the Author)

The authors have satisfactorily addressed the comments from the first round of reviews, largely following all the suggestions and providing additional clarity for non-specialists that is much appreciated. The separation of Figure 1 into new Figures 1 and 2 was particularly welcome. I am now happy to recommend this manuscript for publication.

Reviewer #3

(Remarks to the Author)

Dear authors and editor,

In this second round of reviews, the key concerns I had raised have been answered and the necessary amendment of the manuscript have been addressed. I have no more concerns for the publication of this manuscript in Nature communication - as I said previously, to me it is a very interesting and valuable addition to the field. My suggestion is "accept".

Kind Regards,

Olivier Bernard

Amphibole Reaction Rims Record Shear during Magma Ascent

Dear Reviewers,

We would like to express our many thanks for your expert reviews of this work; the detailed and technical examinations were highly insightful, and we believe implementation of the recommendations has improved the manuscript. We hope that all of the revisions made have appropriately addressed your comments. In addition, the manuscript has now been fully formatted in accordance with *Nature Communications* guidelines; as this submission was transferred from *Nature Geoscience*, which permits Extended Data figures, all Extended Data have been consolidated into a single, correctly formatted Supplementary Information file. Please find below responses to each comment, along with a point-by-point reply, detailing how we edited the manuscript based on your recommendations. All of our responses are in green text.

Sincerely,

Paul A. Wallace and co-authors

Reviewer #1 (Phil Shane)

Summary

Wallace et al present an interesting and convincing investigation into the impact shear deformation on the texture of reaction rims formed during the breakdown of amphiboles during their eruption. It is well known that pressure and temperature changes can quickly result in the breakdown of amphibole when taken outside its stability field, and many workers have used this to infer processes that occurred during historic eruptions. In some cases the texture of the reaction rims has been attributed to decompression and hence ascent rates of the magma can be determined. Such studies are significant in volcanic hazard assessments. In other eruptions, arguments have been made for temperature control on rim formation. A combination of factors was suspected. However, this new work by Wallace et al shows that some of the textural variability is due to shear deformation in the conduit during magma flow, and that the fluid properties can change and potentially be deciphered by micro-imaging. This adds to the potential information on magma ascent that could be inferred from amphibole rim textures. The authors use examples from well-studied volcanic eruptions that are known to show some variation in processes, and compare them to experimentally made crystals. They introduce imaging and crystal orientation techniques that will be useful in future studies.

Hence, the work is noteworthy. Understand the processes of explosive volcanic eruptions is a topic of wide interest. This is an important contribution to the literature on interpreting amphibole rim textures for volcanic phenomena. It is new and compliments the current literature. The conclusions all seem logical from the data presented and the description of the methods is sufficient to be convincing and would act as a basis for using this approach in other studies in the future.

There is sufficient information on the numeric or modelling, but I am not an expert on such calculations. However, it is compelling.

Overall, a welcome contribution, worthy of publication.

We thank Reviewer 1 for this very positive assessment and are pleased that the novelty, rigor and broader impact of our work were clearly appreciated. We have further strengthened the clarity of the modelling description and expanded details of assumptions and limitations in response to Reviewers 1 and 2 (see below).

Minor Comment

Minor text error?: in paragraph 3 and 4 of section named “amphibole breakdown...”, the wrong panels in Fig 1 are referred to. Check.

Thank you for identifying this error. You are correct and this has been changed. We have also checked all other figure references to make sure this has not happened elsewhere.

Reviewer #2

Summary

This paper presents EBSD-derived crystallographic data from amphibole reaction rims, including phase and crystal orientation data, to investigate the influence of deformation on the nature of amphibole breakdown during magma transport. They find that while some reaction rims record systematic topotactic relationships between the amphibole and pyroxene grains, other rims contain systematically reoriented grains. Models are consistent with this reorientation relating to shear deformation during magma ascent.

The authors' consideration of deformation during magma transport certainly poses an important question about how microstructures are interpreted to reconstruct a P-T-t path. The manuscript presents an interesting dataset spanning multiple volcanic systems and compares natural microstructures with models, but the assumptions used in modeling the deformation are not clearly presented. However, since I do not have any expertise in magma transport and evolution, my review is focused on the interpretation of the crystallographic data and the discussion of deformation. The manuscript would benefit from some refinement of the figures and much more clarity about the style of deformation that seems to be implied throughout the text. I think this paper can be a meaningful contribution, and I recommend major revisions.

We thank the reviewer for their careful and thoughtful assessment of our manuscript. We appreciate their recognition of the novelty and significance of linking amphibole reaction rim microstructures to deformation during magma ascent. The detailed comments regarding model assumptions, deformation style, and figure clarity were very helpful and have substantially improved the clarity and robustness of the revised manuscript.

Main Comments

From what I can gather in the supplement, the forward modeling of crystal orientations is assuming that the newly formed crystals are freely rotating as rigid particles within a viscous matrix driven by a velocity field.

The reviewer is correct. The forward modelling framework assumes that newly formed pyroxene crystals in the reaction rim behave as rigid particles embedded in a viscous melt and rotate in response to the local velocity gradient. We do not simulate intracrystalline deformation within the pyroxenes. We have now clarified this assumption explicitly in the main text and Methods. EBSD data show that internal misorientations within rim pyroxenes are generally very small (typically $\leq 3^\circ$), indicating that crystal-plastic deformation is negligible in most rim grains, with the exception of rare crystals in the innermost rim. These small internal misorientations cannot explain the large, systematic orientation distributions observed at the grain population scale.

From first principles, rigid-body rotation of anisotropic particles in viscous flow produces predictable, systematic changes in crystallographic orientation through time. Our numerical approach is based on the

analytical solution of Jeffery (1922), and the resulting orientation evolution is fully consistent with the observed EBSD patterns.

A misorientation is calculated given some initial crystallographic orientation of the amphibole and pyroxene grains.

Yes. In the revised manuscript, we now explicitly state that pyroxenes are initialised with a topotactic orientation relative to the host amphibole, based on the orientation relationships documented in the inner rims. Misorientation is then calculated as the angular deviation between the evolving pyroxene orientation and the fixed crystallographic reference frame of the amphibole host through time. This procedure is now described explicitly in the methods.

Given that the rotations observed from the EBSD data (i.e., the b-axis is the dominant rotation axis), it seems there could be some contribution of intracrystalline deformation to the reorientation of these grains. Could this contribution be significant, or are there any constitutive relationships that are included in the model setup? Perhaps the strain rates are much too fast for this to be under consideration, but with velocities down to 10^{-9} m/s and temperatures of >800 °C, I think it's a possibility. At the very least, the authors do not provide an adequate interpretation for why the rotations they observe are so systematic in line with the crystal directions during the deformation-induced reorientation.

The EBSD data show that the dominant rotation axis of the pyroxenes corresponds to the crystallographic b-axis, which is also the shortest axis of the crystal. This observation supports a rigid-body rotation mechanism, in which anisotropic grains rotate preferentially around their shortest axis during viscous flow. Minor intracrystalline deformation is detectable in some pyroxenes in some inner rims, but is typically $\leq 3^\circ$ (as shown in Supplementary Fig. 4), and therefore cannot account for the large orientation spreads observed. While intracrystalline deformation could, in principle, contribute if sufficiently developed, it is insignificant in reaction rims analysed here.

Alternatively, are the microlites consistently anisotropic grains that are also demonstrating a shape-preferred orientation as their long axes align with the flow direction?

The reviewer is correct. Rigid-body rotation in our model requires the rim crystals to be anisotropic. We provide a histogram of pyroxene aspect ratios in Supplementary Fig. 12, which shows that the grains are predominantly elongate. Equant grains also reorient relative to the shear field but tend to undergo tumbling behaviour, periodically aligning with the flow without stabilising to a strong preferred orientation (Supplementary Fig. 15). This behaviour further supports rigid-body rotation as the primary reorientation mechanism.

Overall, there is a significant lack of information when it comes to the interpretation for a mechanism behind reorientation beyond "deformation." Currently, the main text is not clear about the type of deformation, in terms of common deformation mechanisms, the authors are discussing throughout the manuscript. There is only some clarification in the methods section at the very end of the manuscript. At a minimum, I would ask that some further clarification about the model setup and assumptions is added to the main text of the paper indicating the kind of deformation the authors are modeling.

We have clarified throughout the main text that the deformation modelled corresponds to magma flow-induced strain fields, rather than solid-state deformation. Specific changes include:

Line 25–26: Added "magma flow": "...numerical simulations of crystal rotation under magma flow"

Line 89: Changed “deformation” to “magma flow”.

Line 142: Added “magma flow”: “To assess whether deformation by magma flow could disrupt topotactic alignment...”

Line 143: “we use forward models to simulate crystal reorientation due to rigid-body rotation under strain fields induced by...”

Line 172: We have separated reference 45 to go with “strain field around a settling amphibole crystal” and reference 46 with “rotation of pyroxene microlites”.

I would also recommend adding a table to the supplement or an Extended Data Table 3 that summarizes the model parameters and the values used.

We have added a new Supplementary Table 3 summarising all model input parameters for the settling, lid-driven cavity, and simple shear simulations, including amphibole radius, velocity, time, strain, number of pyroxenes, number of orientations, rim thickness, and aspect ratio statistics. See below.

Input parameters	Value		
	Settling	Lid-driven cavity	Simple shear
Amphibole radius	5×10^{-4} m	$\gg 1 \times 10^{-3}$ m	1×10^{-3} m
Velocity	1×10^{-8} m/s	1×10^{-3} m	1×10^{-3} m
Time	48 hr	50 s	100 s
Strain	3.7	50	100
Number of pyroxenes per orientation	1000	400	1000
Number of orientations	100	1	100
Rim thickness	1.5×10^{-5} m	1×10^{-3} m	3×10^{-5} m
Aspect ratio center	8		
Aspect ratio STD	5		

Minor Comments

Lines 42–43 What kind of deformation is expected (or has been observed) during magma ascent?

During ascent, magma experiences deformation dominated by viscous flow, typically with strong simple-shear gradients due to no-slip conditions at conduit/dyke walls and the development of shear localisation in crystal-rich or degassing magmas. In addition, ascent in brittle crust commonly involves dike propagation, where magma flow is coupled to elastic/brittle host-rock deformation and fracture at the dike tip. In highly crystalline systems and near the viscous–brittle transition, magma may also deform by mixed viscous–brittle processes (e.g., shear-induced microcracking/cataclasis). We now clarify in the manuscript that the deformation discussed refers primarily to flow-induced strain fields (simple shear and localised shear) expected during transport and ascent, rather than solid-state tectonic deformation. This sentence has been updated to be more specific about the deformation we are discussing: “However, the flow-induced strain associated with simple shear and shear localisation during magma transport through dykes and conduits in the crust has rarely been integrated into petrological models”.

Lines 64–67 References for the influence of deformation on reaction kinetics?

We agree that the sentence would be stronger if it explicitly connected “mechanical processes” to reaction kinetics, rather than only to texture. The sentence has been changed to: “However, this P–T–X-centric approach implicitly assumes that breakdown proceeds exclusively through thermodynamic triggers, overlooking mechanical processes or deformation–reaction coupling, despite the fact that shear and flow has been shown to accelerate crystallisation/reaction kinetics and mass transfer in magmas, analogous to well-known strain-enhanced reaction progress in metamorphic systems”.

Lines 136–137 Is this reorientation driven by rigid grain rotation or dislocation creep?

The reorientation we are referring to is rigid-body rotation. The sentence has been updated to clarify this: “In both pyroxenes, [010] (i.e., b-axes) serves as the dominant rotation axis during reorientation by rigid-body rotation.

Lines 163–164 The source of these experiments should include a reference here.

The source of the experiments have been referenced. The sentence has been updated for clarity: “Time-series experiments²², where amphibole grains in contact with a melt were heated 30 °C above their stability field (from 870 °C to 900 °C) and held over durations ranging from 3 to 48 hours, illustrate the reorientation evolution of microlites in the reaction rim (Fig. 2g).”

Lines 563–564 The match files used in Aztec must be reported. “Standard files” is insufficient.

The match files for EBSD have now been reported in the methods section: “Indexing was performed on hornblende, augite, enstatite, anorthite, biotite, magnetite and ilmenite using crystallographic match files with known crystal symmetries⁵⁸, sourced from the American Mineralogist database (using best in database family and 47–50 reflectors)”.

Line 572 The suitability of a minimum of 3 pixels per grain for the grain reconstruction can only be determined from the combination of the step size (0.1–1.0 microns) and the grain size. Please report here what the typical grain size range was for the pyroxene grains.

We now explicitly report the typical grain size range of the pyroxene grains and clarify how EBSD step size was selected relative to grain size in order to ensure robust grain reconstruction.

The grains in the amphibole reaction rims across all samples span a range from approximately 3–20 μm in equivalent diameter. To account for the variability from one amphibole reaction rim to the next, EBSD step size was adaptively chosen based on the expected grain size in each dataset. For rims dominated by very fine-grains, step sizes of 0.1–0.2 μm were used, whereas for rims with coarser grains, step sizes of 0.5–1.0 μm were employed.

With this approach, the majority of reconstructed pyroxene grains were sampled by significantly more than the minimum 3 pixels per grain, and grains reconstructed with the minimum pixel threshold represent only a very small fraction of the total population. We have now added the typical grain size range and corresponding step sizes to the methods section of the manuscript.

Figures Comments

These figures are very dense and not easy to read. Many of the plot labels are much too small. I think most if not all the figures would benefit from adding further differentiation between labels and sublabels as well as more white space between the separate sections.

We thank the reviewer for this helpful suggestion. All figures have been revised to improve readability by increasing white space, enlarging plot labels, and improving the visual distinction between labels and sublabels. In addition, Figure 1 has been split into two separate figures to reduce visual density and improve clarity. We think these changes significantly enhance figure legibility without altering the underlying data or interpretations.

Figure 1 This is a very dense figure. The labels are hard to read when placed over the data. In the caption, the (100), (010), and (001) planes should still include the parentheses indicating that they are planes.

The parentheses indicating crystallographic planes (e.g., (100), (010), (001)) have now been added consistently in the figure caption. To address the issue of visual density and improve label readability, the original Figure 1 has been separated into two figures, with labels repositioned and additional white space introduced. Further details on these changes are provided below.

Additionally, the print resolution of the figure overall, but particularly the orientation maps and pole figures is insufficient for the level of detail, I strongly recommend making sure the final version has a higher dpi.

We thank the reviewer for this important recommendation. All figures have now been regenerated and exported at high resolution (minimum 600 dpi), ensuring that orientation maps and pole figures are shown at a level of detail appropriate for publication. We note that during the initial submission process, figures embedded within the manuscript file were automatically compressed by Microsoft Word. This issue has been addressed in the revised submission to ensure full-resolution figures are retained, and we will ensure full-resolution is available for the publication.

Figure 1a Why does this map only include one complete grain? Assuming this map is made from grain separates, what's the reason for not making complete maps of these 5 grains?

We thank the reviewer for this question. EBSD maps were collected for a limited number of complete amphibole grains because fully mapping entire grains is extremely time-consuming at the step sizes required to resolve fine-grained reaction rims. Mapping complete grains would have substantially reduced the total number of amphiboles that could be analysed and, consequently, the statistical robustness of the dataset.

For the purposes of this study, mapping the entire amphibole grain was not necessary, as the key requirement was to capture a representative section of the reaction rim together with the host amphibole, which provides the crystallographic reference frame for the analysis. All EBSD maps therefore include both rim minerals and the adjacent amphibole, ensuring consistent orientation reference while maximising the number of reaction rims analysed.

And are these 5 grains different from the grains used in Fig. 1c,f,i,l,o?

Yes, the grains shown in Fig. 1a were different from those used in Fig. 1c,f,i,l,o. In the original figure, panel 1a was intended to highlight particularly clear examples of amphibole reaction rims from the four

volcanic systems, illustrating their textural and mineralogical variability. Panels 1c,f,i,l,o were intended to show type examples of crystallographic evolution, ranging from highly topotactic reaction rims observed in experiments and at Unzen, through mixed alignment in Soufrière Hills Volcano samples, to strongly misoriented rims at Bezymianny and El Misti.

We recognise that presenting these elements together in a single figure was confusing and contributed to visual density. To address this and improve clarity, we have now separated the original Fig. 1 into two figures. The revised Fig. 1 now includes panels a and b only, focusing on petrographic context and reaction rim textures, whereas the revised Fig. 2 includes panels c–q, which present the EBSD-based crystallographic analyses. This separation both improves readability and clearly distinguishes between the petrographic observations familiar from previous studies and the crystallographic analyses introduced in this work, while also allowing for increased white space within each figure.

I would also recommend listing the 4 volcanoes in the legend on the right-hand side of Fig. 1a in the same order that they appear in Fig. 1c–q.

Thanks for this recommendation. These labels have now been made clearer without overlapping data.

Figure 1b Please state explicitly in the caption what the amphibole symbols mean in the schematic. Is that depth levels where amphibole was recovered from samples? Are those samples included in the manuscript?

The amphibole symbols in Fig. 1b represent the depths (or pressure levels) within the volcanic system at which amphibole stability and initial crystallisation have been reported, based on previous petrological studies cited in the figure caption. These symbols do not indicate sampling depth within a single eruption deposit, but rather the inferred magma storage depths where amphibole crystallised prior to ascent.

In several volcanic systems, including Unzen and Bezymianny, amphiboles have been reported to crystallise at multiple depths, commonly interpreted as reflecting deeper and shallower magma storage regions. Subsequent magma recharge and mixing can entrain amphiboles from these distinct storage zones, resulting in erupted products that contain multiple amphibole populations. The amphiboles analysed in this study are drawn from these erupted products and therefore include crystals originating from these different storage depths.

We have now clarified this explicitly in the figure caption to avoid confusion.

Figure 1c... Please state in the figure caption what half-width was used for contouring these 1-point-per-grain pole figures.

We have now specified the half-width used for contouring in the figure caption. The contoured pole figures were generated in MTEX using a 10° half-width kernel for the 1-point-per-grain pole figures (one mean orientation per grain).

Figure 2c Please clarify the variables used for plotting strain (u / A) in the caption. And is the orientation of the cartoon amphibole grain representative of the initial orientation? Or is everything relative and the actual orientations aren't significant?

We have clarified in the figure caption that strain is dimensionless, defined as velocity (u) scaled by crystal radius (A), which yields a strain field that depends only on the scaled radius. For panel 2c, the schematic crystal orientation is arbitrary, whereas in panel 2d all orientations are shown in the model reference frame (z -axis downward).

Figure 2a,g The source of the experimental samples should be referenced in the figure caption.

The reference to the experimental paper of De Angelis et al. 2015 has now been included in the figure caption.

Figure 5a Scale bars should not be placed over busy data as they become too difficult to read. I suggest placing a white or transparent box underneath for readability.

A white box has been added behind the scale bar to improve readability.

Reviewer #3 (Olivier Bernard)

Summary

Dear Authors and Editor,

The manuscript entitled “Amphibole Reaction Rims Record Shear during Magma Ascent” by Paul A. Wallace and collaborators, proposes a framework to use amphibole breakdown rims as tools to quantify strain in magmas. With EBSD (Electron backscatter diffraction) measurements on natural data and experimental results to quantify the orientation of amphibole and the crystals composing their breakdown rims they hypothesize that re-orientation of crystals was the result of deformation. To test this, they use flow mechanics simulation of amphibole surrounded by elongate pyroxene exposed to several deformation fields (crystal settling, flow across a flat surface with a notch, simple shear), and calculate the reorientation of pyroxenes over time. They conclude that in the case where Strain rate decreases relative to crystallization rate (development of the rim), the crystal grows following the alignment of the “host” amphibole (topotactic growth), and that with the opposite situation, misalignment with host amphibole increases.

This paper shows a novel and very interesting approach to use amphibole breakdown rims that could help to quantify to some extent magma deformation during its late storage, ascent and eruption. Data from models, petrological experiments and natural samples seem to agree with the argument made as well and represent a quite extensive EBSD dataset.

However, I have some reserves because of the underlying assumptions made during the modelling approach that uses only a single crystal exposed to flow (Stokes flow, simple shear), without considering the influence of multi-crystals, which has been shown to drastically change flow simulation results.

Besides, I would have liked to see more data on the breakdown rims from natural products (Percentage of amphiboles with rims, thickness distribution of the rims for instance) to be able to assess the representativity of the sample chosen for EBSD. The conclusion using the MCMC model is also very interesting, but I think it lacks some of the data that were used to calculate the decompression rates using amphibole breakdown rim models.

Overall, I think this study fits in Nature Communications thanks to its novelty and comprehensive approach, but I think there is still some key points that need to be addressed and discussed for this study to be published (see my more detailed comments below).

We thank the reviewer for their detailed and constructive evaluation of our manuscript. We appreciate their recognition of the novelty and potential of using amphibole breakdown rims to quantify magma deformation, as well as their thoughtful comments on the modelling assumptions and sample representativeness. Their suggestions have helped us to clarify the scope our modelling approach and to strengthen the revised manuscript.

Main comments

1) Single crystal flow models:

In your model about crystal settling, you use a simple case of Stokes law for slow settling velocities. However, above even relatively small crystal volume fractions of a few percent, this type of settling doesn't completely describe the forces applied on the crystals. For all your case studies, you are above this threshold, and therefore other forces applies slowing down crystal settling such as Basset forces, lubrication forces, hindrance and crystal-crystal interactions making the dynamics a lot more complex and the resulting deformation field very different (see Faroughi and Huber 2015, Culha et al., 2020, Huber et al., 2025 Ann Reviews). This could have a big influence on your modelled misorientation outputs. The dynamics of two-phase flow (crystals + melt) can be very complex and upscaling to a whole magma chamber or conduit the behaviour of a single crystal in an infinite fluid volume is a severe simplification.

Thank you for this constructive comment. In the limit where the separation distance between crystals is much larger than the amphibole radius, neighbouring crystals do not affect the local flow field. As long as the separation exceeds the pyroxene length scale, the influence of neighbouring crystals is minimal. When crystals are closer than a few pyroxene lengths, local shielding or shear amplification may occur, but these effects can either reduce or locally enhance deformation. Our modelling focuses on the strain experienced at the amphibole–rim scale, and incorporating full multi-crystal interactions would require numerical approaches beyond the scope of the Monte-Carlo framework used here.

We have added the crystal volume fraction used in the settling experiments (~5 vol.%). At these low crystal fractions, hindered settling effects reduce velocities by ~30%, which is smaller than the uncertainty associated with melt viscosity. Importantly, even if the melt viscosity was perfectly constrained, our simulations are sensitive to accumulated shear strain rather than absolute velocity. Any reduction in settling velocity may be compensated by localisation of shear between crystals. But either way, given the low crystal fractions and goals of the model, the effect is likely not important, and moving from the analytical description of an analytical flow field to a fully numerical two-phase flow simulation is not warranted.

Clustering particularly can occur even at low crystal fractions (Culha et al., 2020), and mean a cluster would behave as a single settling particle – could this also explain the presence of asymmetric rims? I think all these limitations should at least be mentioned and discussed in the manuscript, and maybe some testing could be done on the simpler crystal settling case.

Crystal clustering is particularly important when there are spatial gradients in the volume fraction of crystals. In the suggested paper, which is an excellent example of the state-of-the-art on this topic, the authors find that crystal clustering is only really important when there is a crystal-rich layer above a crystal-poor(er) layer. While this may be relevant in magma chambers, it is not applicable to our experiments, which are seeded homogeneously with amphiboles. Although clustering could in principle generate asymmetric shear fields, the amphiboles we image are well explained by apparent asymmetry related to sectioning geometry/slice direction, and invoking clustering is not required to reproduce the observed textures.

2) Sample representativity:

How representative are amphibole analysed for each eruption?

To ensure representative of the range of reaction rim development within each eruption, amphiboles were deliberately selected to span the full spectrum of observed rim textures in each sample, including thin

rims, thick rims, and nearly completely pseudomorphed grains. This targeted sampling strategy was designed to capture the crystallographic evolution of amphibole breakdown across different stages of reaction progress rather than to quantify population-scale abundances. Importantly, within each eruption, amphiboles with different rim thicknesses exhibit consistent crystallographic misorientation patterns, indicating that the deformation signal extracted from the rims is robust and not dependent on selecting a particular subset of crystals. An extra paragraph has been added to the methods section of the manuscript to clarify representativeness and reason behind the amphibole rims selected.

EBSD is a time-consuming method, so I understand that it is not possible to analyse hundreds of amphibole per eruption deposit, but it would be good to have an idea of amphibole size distribution in each eruption: how many have rims, what is the thickness distribution of these rims, etc (in a similar way that you show on your extra figure 1 for rim mineralogy).

We fully agree with the reviewer that sample representativeness is an important consideration. However, we wish to clarify that the objective of this study is not to characterise population-scale amphibole textures within eruptive deposits, but rather to quantify the crystallographic response of amphibole reaction rims to deformation. For this reason, our sampling strategy was deliberately process-focused rather than population-statistical.

Specifically, we intentionally selected amphiboles that preserve reaction rims, because these are the crystals that record the deformation-controlled microstructural information that is the focus of this study. As most amphibole crystals in volcanic deposits are present as a phenocryst phase (>100 μm diameter), we ensured these were the grains we analysed. This targeted approach is standard practice in amphibole breakdown studies and has been used for more than three decades in rim-based reconstructions of magma ascent rates and thermal/decompression histories. Our aim is therefore not to estimate how common rimmed amphiboles are within each eruption, but to extract the deformation signature recorded by those rims once they are present.

To provide full transparency in our amphibole selection process, we have provided the whole thin section scans for all samples with the targeted amphiboles highlighted in a new Supplementary Fig. 1.

Within each sample, we further ensured representativeness by analysing amphiboles that span the full observed range of rim development for that sample, including thin rims, thick rims, and nearly completely pseudomorphed amphiboles where present (as the parent amphibole acted as the reference orientation framework for our reaction rim pyroxenes, we needed to ensure the targeted amphibole crystals had not fully pseudomorphed). This ensures that our dataset captures the entire spectrum of reaction progress and textural maturity, rather than being biased toward any particular subset (e.g., only thick rims). To demonstrate this, we have added an additional supplementary figure, in a similar manner to the rim mineral figure (Supplementary Fig. 2), that shows the reaction rim thickness distribution for the selected amphibole reaction rims across each sample (see new Supplementary Fig. 3). It is important to note that rim thickness is not directly comparable between different samples or volcanic systems, as it reflects system-specific ascent histories, thermal conditions, and degassing processes; for example, amphibole reaction rims at Soufrière Hills Volcano commonly attain much greater thicknesses than those observed at other volcanoes, and we therefore did not use rim thickness as a cross-system selection criterion in this study.

Importantly, we show that, despite differences in rim thickness, amphiboles from the same eruption display consistent crystallographic misorientation distributions, indicating that the deformation signal we extract is robust and not an artefact of rim size or reaction progress. Additionally, rim thickness does not systematically correlate with crystallographic misorientation, as both thin and thick reaction rims display similarly strong misorientation patterns. For example, amphibole reaction rims from Unzen, Bezymianny, and El Misti exhibit similar thickness distributions, yet Unzen rims preserve a strong topotactic relationship with the host amphibole, whereas those from Bezymianny and El Misti do not (see new Supplementary Fig. 3).

We nevertheless agree that population-scale metrics such as rim abundance and thickness distributions are valuable for broader eruption-scale petrological questions, and we view these as a natural direction for future work building on the framework developed here.

We therefore consider that adding population-scale data would not strengthen the mechanistic conclusions of this study, which are based on the internal crystallographic relationships within individual reaction rims rather than on their frequency in the eruptive products. Despite this, we have added new supplementary figures to support sample representativeness.

Line 222-224: You mention, loss of crystals from the rim because of the strain applied to the amphibole. Did you quantify it? With a given strain, how much of the rim thickness could be lost?

The rate of crystal loss from the rim depends on several factors, including the thickness of the rim, the position of individual crystals within the rim, the shear geometry, and the shear strain. Generally, crystals located further from the amphibole experience a faster loss rate, resulting in an accelerated movement away from the amphibole. This can be observed from the streamlines in the settling geometry, which indicate that rim crystals are advected along the surface of the amphibole. Consequently, only crystals situated at the very top of the rim are lost, and even then, at a relatively slow pace. In the simulations presented, we initiate with an initial rim thickness of $0.03A$. After a total strain of 3.7, only a few crystals (less than 0.1%) exceed a distance of $0.033A$ from the amphibole. Conversely, in the simple shear geometry, crystals are advected parallel or sub-parallel to the far-field flow direction. Crystals that move more than approximately $0.1A$ away from the surface are lost (freely move with the melt phase). Under the same initial rim thickness as the settling case, approximately half of the crystals are lost (defined as being more than $\sim 0.1A$ away) after a strain of ~ 2 , and almost all crystals are lost after a strain of ~ 20 .

It would be nice to have details on the parameters used after the MCMC simulations to obtain decompression rates at each step from the amphibole breakdown rim models.

We now provide the growth-rate parameterisation used to convert MCMC outputs to decompression histories. Growth rates are defined by linear interpolation between:

300 MPa: $0 \mu\text{m}/\text{day}$;

110 MPa: $0 \mu\text{m}/\text{day}$;

105 MPa: $7 \mu\text{m}/\text{day}$;

0 MPa: $0 \mu\text{m}/\text{day}$.

These parameters are now reported explicitly in the Supplementary Information.

Minor Comments

Line 98: it's Fig 1f-o

Corrected.

Line 541-546: inconsistency with the cited study.

Thank you for identifying this citation error. This has now been corrected.